# Biologically Assisted One-Step Synthesis of Electrode Materials for Li-Ion Batteries

**DOI:** 10.3390/microorganisms11030603

**Published:** 2023-02-27

**Authors:** Laura Galezowski, Nadir Recham, Dominique Larcher, Jennyfer Miot, Fériel Skouri-Panet, Hania Ahouari, François Guyot

**Affiliations:** 1Institut de Minéralogie, Physique des Matériaux et Cosmochimie, Sorbonne Université, Muséum National d’Histoire Naturelle, CNRS UMR, 7590, 75005 Paris, France; 2Laboratoire de Réactivité et Chimie des Solides, CNRS UMR 7314, Université de Picardie Jules Verne, 33 Rue Saint Leu, CEDEX 1, 80039 Amiens, France; 3Réseau sur le Stockage Electrochimique de l’Energie (RS2E), FR CNRS, 3459, 80039 Amiens, France; 4Université de Lille, UMR CNRS 8516-LASIRE Laboratoire Avancé de Spectroscopie pour les Intéractions, la Réactivité et l’Environnement, 59655 Villeneuve d’Ascq, France; 5Université de Lille, FR 2638-IMEC-Institut Michel-Eugène Chevreul, 59000 Lille, France; 6Institut Universitaire de France (IUF), 75005 Paris, France

**Keywords:** biomineralization, biofilms, electroactive biofilms, birnessite, manganese oxides, one-pot electrode synthesis

## Abstract

Mn(II)-oxidizing organisms promote the biomineralization of manganese oxides with specific textures, under ambient conditions. Controlling the phases formed and their texture on a larger scale may offer environmentally relevant routes to manganese oxide synthesis, with potential technological applications, for example, for energy storage. In the present study, we sought to use biofilms to promote the formation of electroactive minerals and to control the texture of these biominerals down to the electrode scale (i.e., cm scale). We used the bacterium *Pseudomonas putida* strain MnB1 which can produce manganese oxide in a biofilm. We characterized the biofilm–mineral assembly using a combination of electron microscopy, synchrotron-based X-ray absorption spectroscopy, X-ray diffraction, thermogravimetric analysis and electron paramagnetic resonance spectroscopy. Under optimized conditions of biofilm growth on the surface of current collectors, mineralogical characterizations revealed the formation of several minerals including a slightly crystalline MnOx birnessite. Electrochemical measurements in a half-cell against Li(0) revealed the electrochemical signature of the Mn^4+^/Mn^3+^ redox couple indicating the electroactivity of the biomineralized biofilm without any post-synthesis chemical, physical or thermal treatment. These results provide a better understanding of the properties of biomineralized biofilms and their possible use in designing new routes for one-pot electrode synthesis.

## 1. Introduction

Abundance, low toxicity, low cost and stability in the ambient atmosphere of MnO_2_-based oxides [1] make them particularly attractive as electrode materials for electrochemical systems, such as primary Zn/MnO_2_ cells (Leclanché, alkaline cells, 1.5 volts) [2,3] or Li/MnO_2_ coin-cells (3 volts) [4,5,6,7,8]. Synthesis conditions strongly influence the texture, morphology, particle size and degree of crystallinity of these Mn-oxides [9] with a determinant impact on electrochemical performances. Controlling the structural and textural properties of such electrode materials for batteries is key to reaching optimal electrochemical performances, but it is still challenging.

Birnessite, a phyllomanganate composed of layers of MnO_6_ octahedra with ≈7 Å interlayer spacing, shows the insertion of cations within its interplanar spaces [10,11]. Various methods have been developed to produce birnessite to be used as an electrode material, including sol–gel reactions, calcinations and hydrothermal decomposition, e.g., [12,13,14], sometimes triggered by the use of strongly oxidizing conditions, and generally under acidic conditions. These methods, however, are most of the time non-environmentally friendly and involve multistep synthesis pathways [15]. To bypass these multi-step processes, it is important to develop a simpler birnessite synthesis pathway, to guarantee an optimized morphology and texture of the electrode.

Our goal is to propose an alternative way of electrode material production. To do this, our approach has comprised using microbial biomineralization to synthesize a Mn-oxide-based electrode. Microbial biomineralization proceeding under soft conditions, i.e., at ambient temperature and in aqueous media, offers a possible abatement of the energetic cost for making electrode materials, as already reported for Fe-phosphates [16], Fe-oxides [17], Co_3_O_4_ [18,19], MnO_x_/C [20], MnO_2_ [21,22] and CuO [23].

In these systems, fungi, yeast and bacteria [24,25,26] promote the formation of mineral–organic assemblages, organized down to the nanometer scale (size range 1–100 nm) in a structure providing better electrochemical performances than their bulk counterparts formed under abiotic conditions. All of these previous studies have explored biomineralization in planktonic cultures, i.e., with bacterial cell suspensions in aqueous media, which have produced materials organized on the single-cell bacterial scale, i.e., submicrometer scale. These syntheses have required some post-processing after biomineralization such as heating and adding a polymer binder or conductive carbon to obtain good electrochemical performances. To go further, we want to propose a one-step synthesis method. 

The main challenge of this study was to limit the post-synthesis processing steps and produce an electrode material directly on a collector. In order to do this, we proposed using one of the fundamental properties of bacteria, the biofilm. Biofilms are organized on a large scale and can colonize large areas (centimeters) [27]; we can therefore envisage colonizing a biofilm on a current collector and obtaining a texture on a centimeter scale. Biofilm-associated cells are different from their suspended counterparts by the generation of an extracellular polymeric substance (EPS) matrix, reduced growth rates and the up-and down-regulation of specific genes [27]. Biomineralization can proceed within these biofilms, with the EPS acting as the preferential site for mineral nucleation and growth [28].

By analogy, biomineralization in microbial biofilms applied to electrode material synthesis could thus provide (1) an organic polymeric matrix acting as an analog of polymeric binders, (2) biominerals acting as electrochemically active matter and (3) a template acting as a current collector at the electrode (Figure 1).

Manganese biomineralization results from Mn^2+^ oxidation to Mn^3+^ and/or Mn^4+^, and is often catalyzed by a microbial multicopper oxidase enzyme [29,30,31,32] in the presence of di-oxygen or mediated by reactive oxygen species. Mn oxides formed by biomineralization are generally characterized by a nanometric size and a poorly crystallized structure. In planktonic conditions, the strain *Pseudomonas putida* produces birnessite, a layered manganese oxide [33,34,35]. This strain is also able to form a biofilm attached to stable surfaces and produce Mn oxide minerals [36,37]. The biosynthesis of manganese oxides constitutes a promising opportunity to be explored given controlling Mn-oxides’ structure and texture from the nano to the cm scale [38,39,40].

Here, as a proof of concept, we report a one-pot synthesis of birnessite-based electrodes without any post-processing above 80 °C. This was achieved using a carbon current collector colonized by a biomineralized biofilm of the *Pseudomonas putida* strain MnB1. Manganese oxides produced by the Mn-oxidizing bacteria acted as active matter of the electrode. We measured the kinetics of Mn^2+^ oxidation via *P. putida* in biofilm and determined the nature of the Mn-bearing biominerals formed. The composition of the electrode and the mineral’s structure and texture, as well as organic matter content, were determined using a combination of X-ray diffraction and absorption spectroscopy, electron microscopies (SEM and TEM), electron paramagnetic resonance (EPR) spectroscopy and thermogravimetric analyses coupled to mass spectrometry. Finally, we evaluated the electrochemical reactivity of these electrodes vs. lithium. Our results highlight the possibility of producing an electrochemically active electrode material of battery vs. lithium in a single step and at room temperature, which opens an alternative route for electrode material synthesis under environmentally relevant conditions. 

## 2. Materials and Methods

### 2.1. Bacterial Culture and Production of Biomineralized Mn-Oxides in Biofilm 

All solutions for culture media were sterilized via autoclaving at 121 °C. Biomineralized manganese oxides were produced in cultures of the *Pseudomonas putida* strain MnB1 (ATCC^®^ 23483™). Bacteria were pre-cultured at 30 °C under aerobic conditions using an incubator system with horizontal stirring (120 rpm) overnight in a pH~6.8 growth medium composed of beef extract (3 g·L^−1^), peptone (5 g·L^−1^) and MnSO_4_·H_2_O (50 µM), after inoculation at 1/60 (*v*/*v*) from a stock culture stored at 4 °C. Then, cells were rinsed three times with a 10 mM HEPES solution with NaCl 10 mM (4000× *g*, 10 min), and then transferred in the stationary phase into 75 mL of biomineralization medium at a cell density of 2.3 × 10^8^ cells/mL. The biomineralization medium (pH 7.40) was composed of HEPES (10 mM), (NH_4_)_2_SO_4_ (2 mM), NaCl (0.7 mM) and glucose (1 mM). MnSO_4_·H_2_O (0.2 mM) was added daily. An autoclaved carbon current collector (gas diffusion layer (GDL—Freudenberg ref H23)) of 0.8 cm in diameter and a thickness of 207 µm, previously weighed, was deposited and left floating at the surface of the biomineralization medium, just before bacteria inoculation. We assumed that the biofilm formed at the medium–air interface close to the GDL current collector and the biofilm formed directly at the surface of the GDL current collector had the same characteristics and so it will be named biofilm-MnO_x_ (Figure 2). The use of biofilm formed at the medium–air interface close to the GDL is justified by the need to have a powder without a current collector for some analyses. Biofilm-MnO_x_ experiments were performed for one week in a dark room, to avoid possible reactions with reactive oxygen species [41], at room temperature and without agitation to promote a uniform biofilm formation. Abiotic controls were prepared in the same way but without bacteria inoculation. Experiments were repeated at least ten times.

### 2.2. Kinetics of Manganese Oxide Precipitation with P. putida 

The analyses were performed in triplicate. Manganese concentrations were measured daily via inductively coupled plasma atomic emission spectroscopy (ICP-AES, Cetac ASX-520). The solution was filtered using a 0.22 µm filtration and acidified with HNO_3_ 2% (Suprapur, Sigma Aldrich, Burlington, MA, USA). Analyses were based on the determination of Mn remaining in the solution in the medium culture. 

### 2.3. Sample Preparation for Biofilm Analyses

Biofilm-MnO_x_ was collected via centrifugation (4000× *g*, 10 min) and rinsed 3 times with water. Colonized GDLs were collected and immersed 3 times in water for 10 min. Samples were further dried under air at 80 °C for two hours. Accordingly with characterization results, we suppose that the specific organization of the biomineral is not significantly affected below 80 °C and that all water loss in this temperature range concerns surface water only. 

### 2.4. Abiotic Reference Compounds

Mn-bearing minerals were used as reference compounds for X-ray absorption, electron paramagnetic resonance and electrochemical studies. H_0.5_MnO_2_ and HMnPO_4_·3H_2_O were prepared via precipitation in aqueous solution. HMnPO_4_·3H_2_O was synthesized from MnSO_4_ 1.275 g and KH_2_PO_4_ 1 g in 15 mL water; the solution was transferred into an autoclave (Parr^®^ 23 mL). Thereafter, the solution was heated overnight at 160 °C. Manganese oxide (H_0.5_MnO_2_) was prepared following the modified method from Villalobos et al. [33]:1 g KMnO_4_ in 100 mL MQ water (Solution (1));2.4 g MnCl_2_·4H_2_O in 100 mL MQ water (Solution (2));0.7 g NaOH in 100 mL MQ water (Solution (3)).

Solution (1) was added slowly to Solution (3) during stirring. Solution (2) was then added to the previous mixture and maintained during stirring overnight. The resulting precipitate was washed 3 times with 50 mL water. All powders were then washed twice with 50 mL water and once with acetone and finally dried in air at 80 °C. MnSO_4_·H_2_O, Mn_2_O_3_ and δ-MnO_2_ are commercial compounds purchased from Sigma-Aldrich. 

### 2.5. Biomineral Characterization

The mass of biofilm formed at the surface of the current collectors was measured by weighing the collector before and after colonization with a microbalance with an accuracy of up to a tenth of a microgram (XP2U Ultra Micro Balance Mettler Toledo^®^, Greifensee, Swiss). The average mass of biofilm on the support was 0.675 mg. 

Proportions of mineral and organic matter in biofilm-MnO_x_ were measured via thermogravimetric analysis (TGA) using an STA449C coupled to a quadrupole mass spectrometer (QMS 403 Aeolos) under dry air flow (50 mL·min^−1^) with a ramp of 5 °C·min^−1^, up to 900 °C. These analyses were coupled with differential scanning calorimetry (DSC). 

Biofilm-MnO_x_ was characterized via X-ray diffraction (XRD) performed in capillaries with a Rigaku MM007HF X-ray diffractometer using a rotating molybdenum anode and a RAXIS4++ imaging plate detector. The background was removed using the software Match!

In addition, bulk Mn speciation was determined via quick X-ray absorption spectroscopy (XAS) at the Mn K-edge (6539–6580 eV) at the ROCK beamline (SOLEIL synchrotron, Saint-Aubin, France). Samples were analyzed at ambient temperature in transmission mode using a Si(111) double-crystal monochromator. Spectra were calibrated by setting the first inflection point of a Mn foil to 6539 eV recorded in double transmission. XAS spectra were merged and normalized and extended x-ray absorption fine structure (EXAFS) data were extracted using the Athena software [42]. X-ray absorption near edge structure (XANES) and *k^3^*-weighted EXAFS data were analyzed using the linear combination fit (LCF) procedure in Athena and a custom-built software based on the Levenberg–Marquardt minimization algorithm, respectively, as described in [43,44]. 

The textures of the biofilm-MnO_x_ were investigated using a field emission gun (FEG) and ZEISS Ultra 55 scanning electron microscopy (SEM) (Zeiss, Marly-le-Roi, France), equipped with an X-ray energy dispersive spectroscopy (XEDS) probe (Bruker). Samples were imaged in back-scattered electron mode at 15 kV (working distance of 7.5 mm) and in secondary electron mode at 3 kV (working distance of 3 mm). 

In addition, biofilm-MnO_x_ was analyzed using scanning transmission electron microscopy (STEM) in high-angle annular dark field (HAADF) mode, via high-resolution TEM (HRTEM) and XEDS, using a JEOL2100F FEG-TEM (JEOL, France) operating at 200 kV. The selected area electron diffraction (SAED) patterns were obtained in the areas of interest and used to characterize crystalline mineral phases. 

EPR spectroscopy (electron paramagnetic resonance) was used to check the presence of paramagnetic species in the samples. Continuous wave (CW) measurements were performed with an X-band Bruker Elexsys E580 spectrometer operating at 9.6 GHz at room temperature, with a modulation frequency of 100 kHz, modulation amplitude of 2 gauss and 2 mW (for powders) or 10 mW (for solutions) of microwave power. Before EPR analyses, powder samples (H_0.5_MnO_2_, planktonic-MnO_2_ [22] and biofilm-MnO_x_) were filled into 3 mm quartz tubes, whereas 2 mm quartz tubes were used to analyze the liquid samples. 

### 2.6. Electrochemical Characterizations in Li Half-Cells

Electrochemical analyses were conducted in laboratory Swagelok-type cells. The active material (AM) used as a positive electrode consisted of biominerals formed within the biofilm nominally assigned to the MnO_x_ composition. The proportion of minerals obtained via the deduction of the mineral part using TGA was used to calculate the mass of active material (AM) in the electrode. Biofilm-MnO_x_ was provided directly from the positive electrodes, without carbon addition. The cells were assembled in an argon-filled glovebox using biofilm-MnO_x_ as the positive electrode separated from the negative electrode (lithium disk) by 2 sheets of glass fiber disks (Whatman GF/D borosilicate), with the whole setup being soaked in an LP30 electrolyte (LiPF_6_; 1 M) solution of ethylene carbonate (EC)/dimethyl carbonate (DMC) mixture (1/1 *w*/*w*). Galvanostatic cycling tests were conducted at room temperature between 2 V and 3.9 V potential at the rate of C/20 (1 electron exchanged per 20 h) or C/50 (1 electron exchanged per 50 h) using a MacPile controller (Claix, France). Specific capacities are reported in mAh per gram of AM (MnO_x_). H_0.5_MnO_2_ was also tested in the Li half-cell. Experiments were repeated at least three times. The proportion of the biomineralized biofilm at the surface of the current collector was low compared to the amount of amorphous material (current collector) and prevented us from being able to analyze the material obtained after the electrochemical cycle. 

## 3. Results

### 3.1. Production of the Electrode: Biomineralized Biofilm Formed by Pseudomonas putida 

*Pseudomonas putida* MnB1 has been reported to produce manganese oxide particles, mainly under planktonic conditions [33,36,45]. Its ability to form biofilms has otherwise been described in several publications [46,47,48]. However, in previous manganese oxide production studies, the medium was stirred, making the production of biofilms on a large scale impossible. In our study, we used this bacterial strain to form a biomineralized biofilm at the surface of a current collector in suitable conditions to produce electrochemically active manganese oxide. 

The synthesis consisted of the incubation of *P. putida* MnB1 in a mineralization medium (Table 1), with a carbon current collector (GDL) placed at the surface of the medium. Without medium agitation, a biofilm developed after 10 h of incubation at the interface between the liquid medium and air as an orange-yellow slime. With daily additions of Mn^2+^, the color of the biofilm turned to a brown tint after 10 h, highlighting manganese oxide precipitation and forming the material Biofilm-MnO_x_ (Figure 2). No precipitation could be observed by the naked eye in the abiotic control in the timeframe of the experiment (2 weeks). Such an air–liquid interfacial biofilm had not been previously described for the strain *Pseudomonas putida* MnB1 but is quite common among bacteria and could be impacted by modifications of pH, the viscosity of the medium and surface tension [49]. The precipitation yields of manganese ranged between 50% and 80% of the total Mn^2+^ added (Figure 3). Chemical analyses indicated that between 1200 µM and 1500 µM out of the total added Mn^2+^ (1900 µM over two weeks) had been precipitated by the end of the experiment (Figure 3). The precipitation of manganese-bearing solids occurred only within the biofilm, and more specifically in the uppermost parts in contact with di-oxygen. Experimentations with medium agitation reported a more efficient Mn precipitation than in this non-agitated biofilm-forming configuration [22,46], probably because Mn-oxidation catalyzed by a multicopper oxidase is dependent on the supply of di-oxygen [31]. The lesser efficiency of Mn^2+^ oxidation in a biofilm compared to planktonic conditions [22,46] is thus likely related to less efficient contact between Mn^2+^ and di-oxygen because of the absence of agitation which is nevertheless necessary to optimize biofilm formation. 

In summary, we can cultivate *Pseudomonas putida* MnB1 without the agitation of the medium to form a biofilm at the interface between the liquid medium and air. The addition of Mn^2+^ into the medium during the process of biomineralization impacted the color of the biofilm and translated Mn^2+^ oxidation into the biofilm. The addition of support (current collector) at the surface of the medium did not prevent the process of biofilm formation. 

### 3.2. Characterization of the Electrode Biofilm-MnOx

Mn-bearing solids produced in Biofilm-MnO_x_ were analyzed using X-ray diffraction (Figure 4). Collected diffractograms display a peak at 7.4 Å, characteristic of birnessite (JCPDS #00-018-0802 [33,34]). In addition, β-Mn^III^OOH, a Mn(III) oxyhydroxide (feitknechtite), was detected in the diffractograms by a shoulder at 4.6 Å [50] (Figure 4). The most plausible way to explain the presence of feitknechtite (β-Mn^III^OOH) is an abiotic reaction between MnO_x_ birnessite and the residual Mn^2+^ not oxidized by the bacteria [11]. This observation confirms that Mn oxides are highly reactive [51]. 

XANES spectra at the Mn K-edge were best fit by a linear combination of Mn^4+^ (52% ± 3%; reference d-MnO_2_), Mn^3+^ (20% ± 4%, reference Mn_2_O_3_) and two contributions of Mn^2+^, corresponding to two distinct references HMnPO_4_·3H_2_O (10% ± 9%) and MnSO_4_ (18% ± 7%) (Figure 5). The addition of two phases of Mn^2+^ improves the fit. XANES spectra results are consistent with XRD data, which showed the detection of MnO_2_, feitknechtite and HMnPO_4_ phases.

Thermal analyses (TGA/DTG) of Biofilm-MnO_x_ were performed to determine the proportions of organic matter (bacterial cells + EPS) and minerals in the electrode material. The thermograms show three distinct phenomena, taking place upon heating under air (Figure 6a,b). The two weight losses in the temperature range of 20–200 °C denote the release of H_2_O coordinated to the interlayer cations, mineral water structure and surface humidity as described by [52]. The amount of water in the biomineralized biofilm measured via TGA was ~10%. The subsequent massive release of water and CO_2_ (~30%) at higher temperatures (ca 200–300 °C) matches the combustion of bacterial organic matter as previously reported in [16]. XRD analysis (data not shown here) indicated that the material collected after TGA was composed of Mn_2_O_3_ bixbyite (JCPDS #96-900-7521). The weight loss at around 550 °C could thus correspond to a reduction in the initial material (Mn^4+^) into Mn_2_O_3_ (Mn^3+^). The absence of any XRD-identified manganese phosphate does not rule out the presence of Mn-phosphate in this material as observed by XANES (Figure 5) but is indicative of its poor crystallinity. 

The total weight obtained after TGA has been used in the calculation of the mineral/organics proportion. Taking into account the mass loss associated with the mineral during TGA, the mineral part in the biofilm (corresponding to a large part of MnO_x_ but also including β-MnOOH and HMnPO_4_·3H_2_O) was ~57 wt% (from a total weight loss of ~43 wt%). This estimation will be used to calculate the percentage of active material (AM) in the electrode. In a first approximation, it was considered that AM has a MnO_2_ stoichiometry. It was then checked a posteriori that taking into account other phases present in the material (see results below) had no significant effect on the calculated proportions.

The current collector (GDL), which constitutes the substrate of the biofilm, was composed of an 8-µm diameter carbon fiber network (Figure 7a). After incubation in a mineralization medium with bacteria, we observed the colonization of the surface of the current collector by the biofilm, further confirmed by SEM observations (Figure 7b). 

Carbon fibers of the current collector were still visible after colonization by the biofilm; bacteria were able to colonize fibers mostly at the surface of the current collector (Figure 7c). Mn oxides were localized in patchy zones of biofilm close to bacteria cells (Figure 7c,d). Aside from dominant carbon (composing the GDL substrate), the biofilm material contained phosphorus, manganese, nitrogen and sulfur (Figure 8b). Mn and P were detected as colocalized by EDX at some places in the biofilm (Figure 8a), whereas Mn-rich zones devoid of P were detected at a distance from the cells (Figure 8c). Manganese oxide agglomerates were porous and consisted of micrometer-sized polydisperse aggregates of nanoparticles (Figure 7d). The formation of a patchy mineral on the biofilm might be related to a heterogeneous distribution of bacteria in the biofilm. Regions hosting more abundant bacteria may have served as preferential Mn-oxidation sites, with EPS locally acting as nucleation points of the mineral. 

To obtain more precise insights into the structure of Biofilm-MnO_x_ down to the nanometer scale, we carried out TEM observations. As shown in Figure 7, the birnessite phase was extracellular (Figure 9a,c,e) and the SAED patterns (Figure 9a insert) confirmed its MnO_x_—birnessite structure. Organic matter, most probably the EPS, is co-localized with MnO_x_ birnessite (Figure 9a). Co-localizations of Mn and P were identified near organic matter or cells (Figure 9c,d) and could correspond to Mn and P adsorbed at the surface of the organic matter or to a Mn(II)-phosphate phase as shown by SEM (Figure 8a) and XAS (Figure 5). In addition, the hydrolysis of polyphosphate granules such as those observed in some cells (Figure 9c) could provide a source of inorganic phosphate that may react with Mn^2+^ sorbed at the cell surface, leading to manganese phosphate precipitation near cells [36]. Mn^2+^ may thus be adsorbed at the edges or incorporated within the structure of birnessite [53,54]. As suggested by XAS (Figure 5), two types of Mn^2+^ are present, one in the form of Mn(II)-phosphate and another absorbed at the surface or in the structure of MnO_x_.

To confirm the presence of Mn^2+^, electron paramagnetic resonance (EPR) spectroscopy is a very useful tool used for the detection of paramagnetic species such as Mn^2+^ or Mn^4+^. The EPR spectrum of Biofilm-MnO_x_ (black curve, Figure 10) consists of one broad peak with 522 G linewidth centered on g ≈ 1.997, on which a well-defined sextuplet is superimposed, indicating that the presence of Mn^2+^ ions in distorted octahedral coordination [55] resulted from hyperfine interactions between electron spin (S = 5/2) and nucleus spin (I = 5/2) [55] (Figure 10). Based on the g value and the narrow linewidth, the EPR signal was assigned to Mn^4+^ (S = 3/2) paramagnetic species which corroborates the findings reported in the literature by Kim et al. on the MnO_2_ biogenic oxide obtained with different bacteria (*Bacillus* spore, SG-1; *Pseudomonas*, GB-1, etc.) [56]. The same features (Mn^4+^ signal) had also been mentioned in the literature in layered Li[Mg_0.5−x_Ni_x_Mn_0.5_]O_2_ oxides [57], in spinels (Li_1+x_Mn_2−x_O_4_ with 0 ≤ x ≤ 0.1 [58], Li_4_Mn_5_O_12_ [59]) and in MnO_2_ powder [60]. Associated with Mn^4+^, the Mn^2+^ sextuplet has also been previously observed in the literature. For example, Kakazey et al. attributed such a sextuplet in MnO_2_ to the presence of traces of the salt initially used for the synthesis associated with the high content of physically adsorbed water at the surface of MnO_2_ particles and in the pores of their aggregates [60]. Another explanation could be provided by the presence of structural Mn^2+^ in the sample as already reported by Najafour et al. [61]. 

The presence of Mn^2+^ in Biofilm-MnO_x_ could also be assigned to Mn^2+^ precipitated as manganese phosphate, Mn^2+^ adsorbed on the material’s surface or Mn^2+^ in the interlayers of birnessite to compensate for negative layer charges [10,62].

The EPR spectrum of the planktonic-MnO_2_ (red curve, Figure 10) displays a broad peak assigned to Mn^4+^ overlapped by a less pronounced sextuplet, indicating the presence of Mn^2+^ as confirmed by the pseudomodulation curve (inset, Figure 10). The Mn^2+^ detected with the planktonic-MnO_2_ could be attributed to a precipitated HMnPO_4_ as already shown by Galezowski et al. [22]; this amount of Mn^2+^ is very low compared to that observed with MnO_x_ biofilm. 

In the case of the abiotic H_0.5_MnO_2_, a very broad signal was obtained (blue curve, Figure 10) attributed to Mn^4+^ without any traces of Mn^2+^ as confirmed by the pseudomodulation EPR curve (inset, Figure 10, red curve). A very broad EPR signal had been obtained by Kim et al., with different synthetic MnO_2_ with a linewidth comprised between 1000 gauss and 2600 gauss [56]. 

In summary, the material produced (Biofilm-MnO_x_) through one-pot synthesis via the biofilm colonization of a GDL current collector and subsequent biomineralization was found to be a composite material. The biofilm and the biomineral, making up this composite material, are intimately engulfed and fixed at the surface of the current collector thanks to the exceptional adhesion capacity of biofilms [63]. It contains MnO_x_ birnessite, β-Mn(III)OOH, Mn(II)-phosphate and Mn^2+^ absorbed at the surface of MnO_x_ birnessite or located in the structure. β-Mn(III)OOH and Mn(II)-phosphate have no electrochemical activity and constitute dead masses in this range of potential (1.9 V to 4 V) [64,65]. As well as an equivalent amount of organic material derived from the cells, EPSs were also detected via TEM analyses, thus making this material a typical organic/inorganic hybrid composite. 

### 3.3. Electrochemical Analysis of Biofilm-MnO_x_

MnO_2_ birnessite commonly studied as an electrode material of Li-ion batteries provides capacities more or less close to the theoretical capacity (308 mAh·g^−1^) depending on the size and the morphology of the particles and the cycling conditions [6,66,67]. Significant research has been devoted to the synthesis of MnO_2_ nanotubes and the development of methods to produce MnO_2_ with a controlled morphology is in demand [68]. 

The electrochemical performances of our bio-assisted electrode (Biofilm-MnO_x_) were evaluated vs. Li^+^/Li^0^ in the conditions listed previously. The potential versus composition profile for a biofilm-MnO_x_ electrode cell is shown in Figure 11a. It reveals electrochemical activity at about ~3 volts vs. Li^+^/Li^0^ in agreement with an active Mn^4+^/Mn^3+^ redox couple [22,69] with a reversible capacity comprised between 130 and 160 mAh·g^−1^ at a C/20 rate (Figure 11a). This capacity corresponds to ~50% of the theoretical capacity of MnO_2_. 

During the first discharge, the capacity corresponded to 80 mAh·g^−1^ at a C/20 rate (Figure 11a) and after the first charge the measured capacity corresponded to 130 mAh·g^−1^, which indicates an extra charging capacity of about 50 mAh·g^−1^. During the second discharge, a capacity near 105 mAh/g was recorded which indicates that the major part of the extra capacity observed in the first charge was reversible. Galvanostatic cycling had been performed in the charge (data not shown here) and the results showed the same extra capacity phenomenon. This reversible extra capacity can be explained if it is attributed to the extraction of a mobile charge-carrying cation that was already in the structure.

Given the mineralized medium solution, mobile extra cations in such structures could be Na^+^ or K^+^, but EDX analyses of biofilm-MnO_x_ showed near-zero cation/Mn ratios (Figure 7c). This could then suggest that H^+^ or Mn^2+^ cations could be present in this biogenic birnessite. 

To obtain further insight into this extra capacity observed during electrochemical cycling, galvanostatic tests of an abiotic H_0.5_MnO_2_ [33] were performed. As displayed in Figure 11e, the electrochemical test vs. Li^+^/Li^0^ revealed electrochemical activity at about ~3 volts vs. Li^+^/Li^0^ and a reversible capacity of 160 mAh·g^−1^ at a C/20 rate. No extra capacity had been observed upon cycling of this abiotic control and this finding could be explained by the absence of mobile H^+^ in the structure. 

The absence of protons’ mobility in the structure of MnO_2_ supports our hypothesis that Mn^2+^ present in MnO_x_ could be removed during the first charge, explaining the extra capacity.

In comparison, the biogenic manganese oxide (planktonic-MnO_2_) obtained with the bacteria *Pseudomonas putida* strain MnB1 in planktonic conditions, i.e., agitated suspension of bacteria and tested in battery vs. Li^+^/Li^0^ in a previous study [22], is shown (Figure 11c). The organic–inorganic composite material showed good electrochemical performances with high power and 300 mAh·g^−1^ of capacity (Figure 11c) and no extra capacity. EPR analysis of this biogenic manganese oxide (red curve, Figure 10) presented only a small amount of Mn^2+^ corresponding to HMn(II)PO_4_, which is a dead mass. This result indicated a low amount of Mn^2+^ in planktonic conditions as also detected by EPR analysis and confirmed the absence of Mn^2+^ associated with the manganese oxides. Moreover, in agreement with this result, galvanostatic tests indicated the absence of extra capacity under planktonic conditions and can thus lead one to conclude on the relationship between the presence of a large amount of Mn^2+^ associated with MnO_x_ in the biofilm and the presence of extra capacity. We hypothesize that Mn^2+^ cations are present in the structure of biofilm-MnO_x_ as detected using EPR spectroscopy (black curve, Figure 10) and XAS analysis (Figure 5) and were removed during the first charge, which could lead to the observed extra capacity. In the case of birnessite with hexagonal layer symmetry, Mn^2+^ or Mn^3+^ have indeed been described with vacancies in manganese–oxygen layers [10,53]

Moreover, this biofilm-MnO_x_ electrode presented a rapid loss of capacity after a few cycles (Figure 11b). In fact, after the fourth cycle, the capacity of the electrode was equal to half of the starting capacity, which reflects low cyclability. 

In comparison, the planktonic-MnO_2_ material presented good cyclability even after 20 cycles (Figure 11d). We note that an absence of extra capacity and good cyclability of the electrode for this condition might be correlated. Thus, the loss of cyclability observed with the biofilm-MnO_x_ electrode could be due to the extraction of Mn^2+^ accompanied by a loss in structure of the active matter. This extracted Mn^2+^ could have been transferred to the negative electrode (lithium disk) and led to the positive electrode’s degradation. 

To verify this hypothesis, an EPR spectrum of the negative electrode (lithium disk) after cycling was collected to detect possible Mn^2+^ traces. For this purpose, electrochemical cycling was conducted with the biofilm-MnO_x_ as the positive electrode and the lithium disk as the negative electrode; after one cycle, the lithium disk was recovered, rinsed two times in DMC (dimethyl carbonate) and then incubated in water. The resulting solution was then analyzed via EPR spectroscopy using a 2 mm quartz tube. The obtained EPR spectrum (Figure 12) exhibited a sextet of lines due to the presence of Mn^2+^ in the solution, which could confirm the hypothesis of Mn^2+^ transfer from the cathode to the anode side. This phenomenon could be responsible for electrode modification during the electrochemical tests, resulting in a loss of cyclability when the biofilm-MnO_x_ was used as the positive electrode (Figure 11b). This feature could also be related to a degradation of the positive electrode in the biofilm case due to structural modifications induced by the extraction of Mn^2+^ during the first charge. 

In summary, we can produce an electrode material for batteries with a biogenic manganese oxide formed in biomineralized biofilm at the surface of the current collector (biofilm-MnO_x_). In the literature, all of these studies require post-synthesis processing to obtain an electrode material. For example, Miot et al. [17] provide the production of α-Fe_2_O_3_ via biomineralization for use as a conversion-based electrode material in lithium batteries. Shim et al. [18] propose the production of porous Co_3_O_4_ that exhibits excellent electrochemical performance in Li-ion batteries. Our method can produce an electrode material in a single synthesis. Electrochemical analyses have confirmed the presence of labile cations resulting in extra capacity during the charge. This presence of labile cations prevents our material from maintaining a good charge after several cycles. 

## 4. Conclusions

The production of an electrode material for Li-ions batteries in a one-step process was achieved through biofilm biomineralization promoted by *Pseudomonas putida* using a GDL carbon current collector. The use of biomineralization in biofilm was interesting for two main reasons: (1) *P. putida*’s ability to produce active matter in a polymeric binder network (the extra-cellular polymeric substances (EPSs) produced by the cells) and (2) the possibility to provide a useful birnessite texture on the electrode scale. 

The characterization of this electrode material revealed a complex composite material composed, almost equally, of both organic matter (bacteria cells and the EPS) and biogenic minerals (MnO_x_, β-MnOOH and HMnPO_4_·3H_2_O). Biogenic minerals, consisting of phyllomanganate birnessite, were highly textured and embedded in the matrix of biofilm, whereas Mn(II)-phosphates were formed in contact with cells and organic matter. Further EPR analyses were performed to obtain more information about the nature and behavior of Mn^2+^ cations present in the structure and indicated the presence of Mn^2+^ related to the electroactive mineral. This electrode material presented electrochemical activity vs. Li^+^/Li^0^ without any post-synthesis treatment. 

Electrochemical tests of this biomineralized biofilm electrode (biofilm-MnO_x_) confirmed the presence of labile cations resulting in extra capacity during the charge. The absence of the extra capacity of both planktonic-MnO_2_ and abiotic H_0.5_MnO_2_ is, according to EPR analysis, related to the absence or low amount of Mn^2+^ in these special electrodes contrary to those of biofilm-MnO_x_, leading one to think that the extra capacity was due to Mn^2+^ in the structure of biofilm-MnO_x_. This extraction of Mn^2+^ could lead to a modification in mineral structure and a modification in the positive electrode, leading to a rapid loss of capacity during the cycles and making the electrode unstable in the battery. 

The presence of Mn^2+^ associated with MnO_x_ could be due to a diminution of enzymatic oxidation because of the absence of agitation in the medium. The Mn^2+^ in the medium could be absorbed and intercalated between layers of lamellar manganese oxides [10]. In addition, we evidenced the impact of the absence of agitation in the medium during the formation of the biofilm. Biomineralization under less oxygenated conditions led to the formation of Mn^3+^-containing phases, which are dead material and negatively impact electrochemical performance. This phase results from the competition between the enzymatic oxidation of Mn^2+^ into MnO_2_ and abiotic reactions between soluble Mn^2+^ and already-formed MnO_2_. 

Our method, using a one-pot production at room temperature with a biofilm, leads to improved electrochemical performances of a birnessite-based electrode. In future work, the challenge would be to optimize the MnO_2_/organic proportions and Mn^2+^ biological oxidation efficiency in order to decrease Mn^2+^ remnants in the final material so as to obtain a more efficient electrode material without the presence of extra capacity. The possibility to obtain an optimized texture of the electrode material without extra capacity could further ensure a better exchange with lithium ions. One solution to optimize the performance of this material could be to use voltage polarization of the current collector during biofilm formation. Polarization can have a favorable effect on the colonization of the biofilm with a supportive or a repulsive effect depending on the bacterial strain, the hydrophilic or hydrophobic nature of the support or the different ionic forces at play in the culture medium [70,71]. The possibility of optimizing the colonization of biofilm on the current collector with an improved part of active matter could be an efficient way of producing manganese oxide. The facility to obtain microscale-textured manganese oxides directly on the current collector could pave the way for the more rational production of electrode materials. 

## Figures and Tables

**Figure 1 microorganisms-11-00603-f001:**
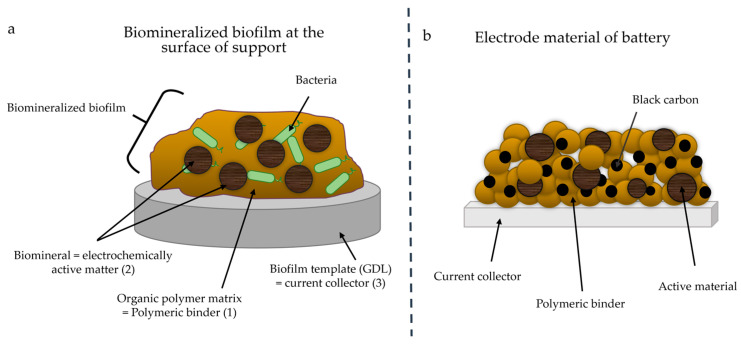
(**a**) Scheme of biomineralized biofilm at the surface of the support and (**b**) of the electrode material of a battery. Each component of the biofilm is analogous to a component of battery electrode material. Black carbon was not added to the biofilm.

**Figure 2 microorganisms-11-00603-f002:**
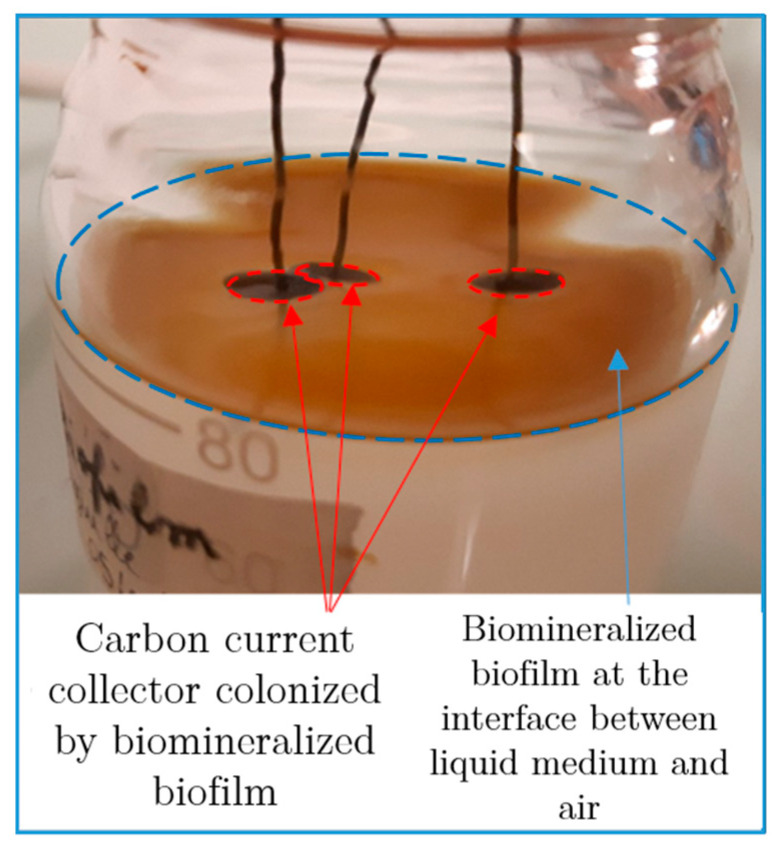
Formation of brown biofilm at the surface of the medium and biofilm colonization on the black current collector after daily additions of Mn^2+^ in the medium.

**Figure 3 microorganisms-11-00603-f003:**
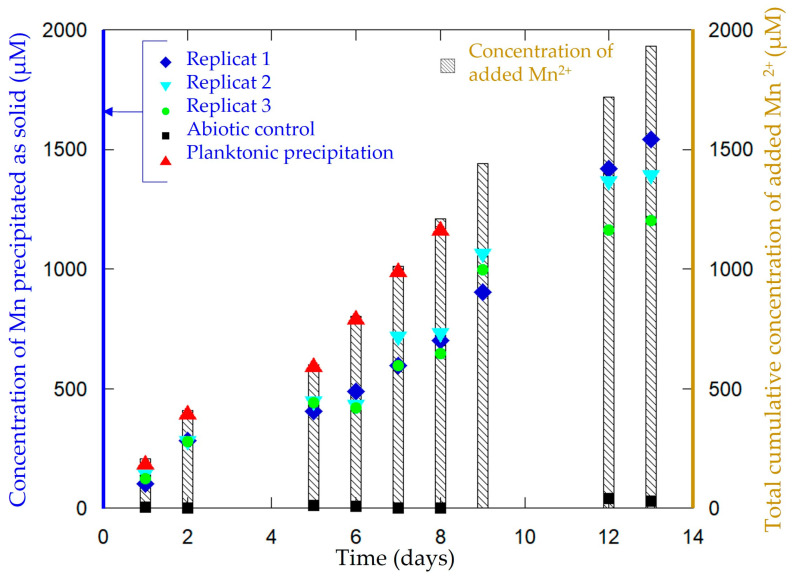
Evolution of the concentration of Mn precipitated as a solid in Biofilm-MnO_x_ (blue squares, indigo triangles and green circles correspond to three replicates in three independent cultures). Total cumulative added Mn^2+^ concentration (bars) increases with time following daily additions of Mn^2+^. Abiotic control (black square) does not show any significant Mn precipitation even after 13 days. In comparison, planktonic control (red triangle) indicates a Mn precipitation rate between 95 and 99% all over the experiment [22].

**Figure 4 microorganisms-11-00603-f004:**
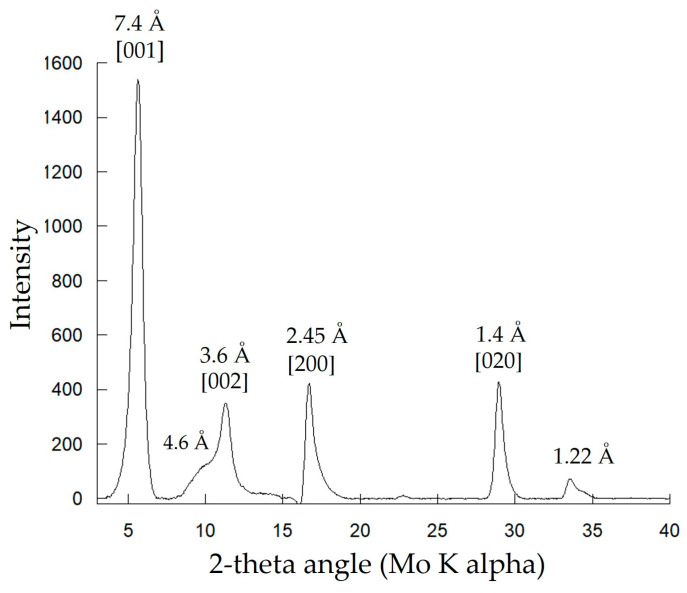
XRD analysis of the biogenic mineral of Biofilm-MnO_x_. Indexations correspond to birnessite except for the peak at 4.6 Å corresponding to β-MnOOH.

**Figure 5 microorganisms-11-00603-f005:**
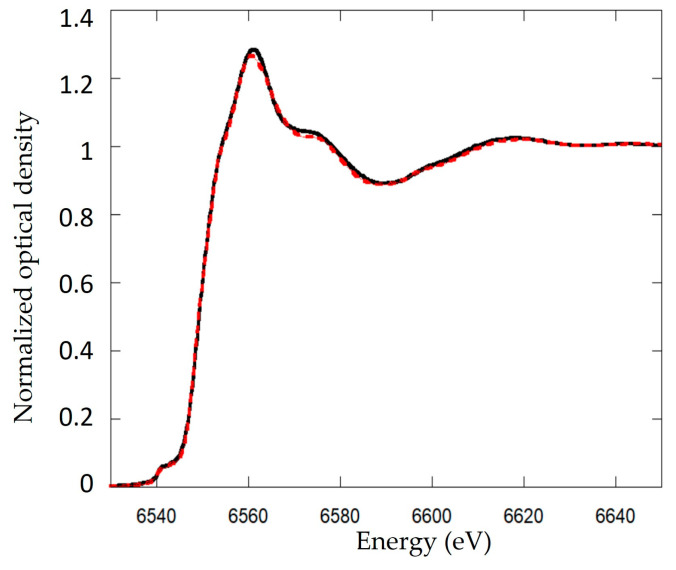
Linear combination fitting (LCF) of Mn K-edge spectra of Biofilm-MnO_x_. XANES spectra at the Mn K-edge compared with the best fits corresponding to a linear combination of 52% ± 3% δ-MnO_2_, 20% ± 4% Mn^3+^, 18% ± 7% Mn^2+^ and 10% ± 9% HMnPO_4_·3H_2_O. The black solid line shows the data and the red dashed line is the best fit of the XANES data.

**Figure 6 microorganisms-11-00603-f006:**
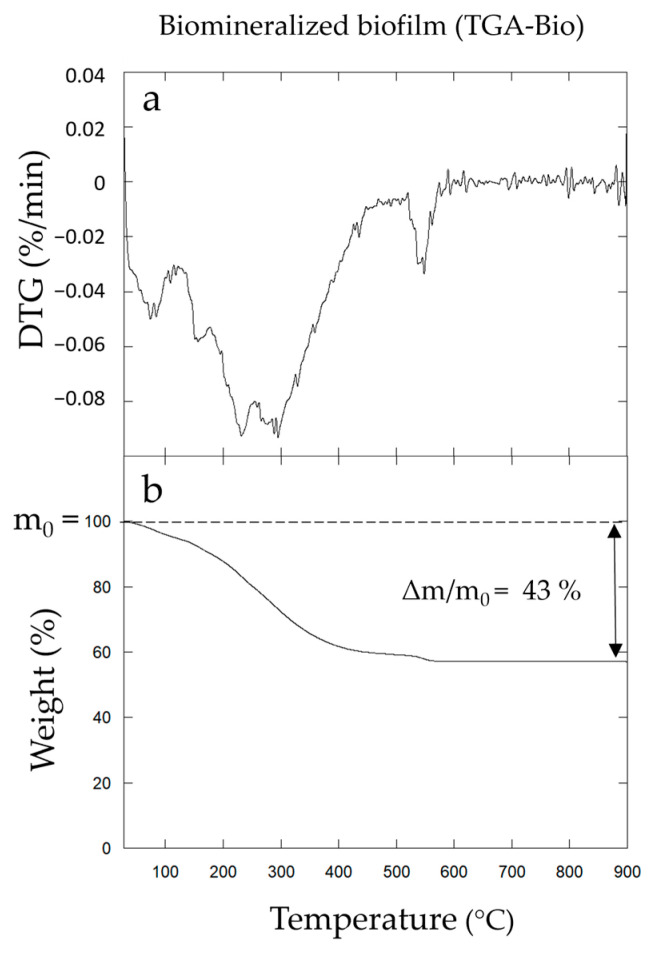
Differential thermogravimetric curves (DTG) (**a**) and thermogravimetric analyses (TGAs) (**b**) of Biofilm-MnO_x_.

**Figure 7 microorganisms-11-00603-f007:**
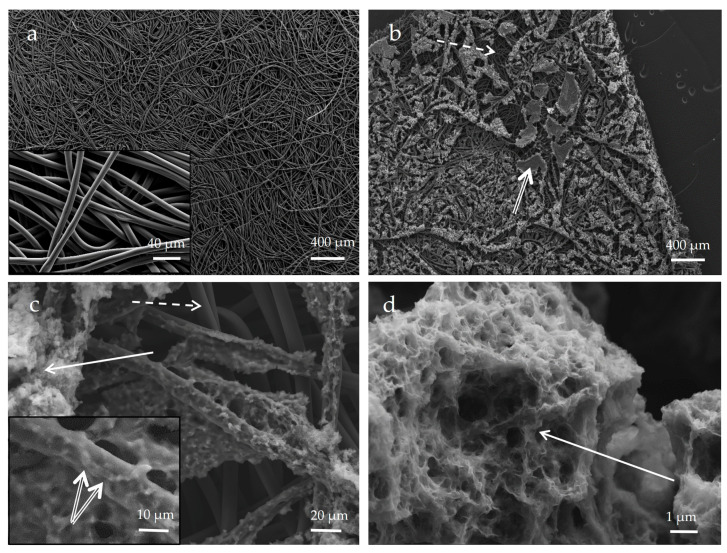
SEM image (secondary electron detector) of carbon current collector before (**a**) and after (**b**–**d**) incubation in biomineralization medium with bacteria (corresponding to Biofilm-MnO_x_). Bacteria in the biofilm (double arrow) are visible (insert in (**c**)) on fibers of the current collector (dotted arrow), mostly at the surface of the current collector; carbon fibers deeper in the current collector are not colonized by the biofilm (dotted arrow). The biomineral formed in the biofilm is close to bacteria cells (solid arrow) (**c**,**d**). Biominerals are porous and textured (**d**).

**Figure 8 microorganisms-11-00603-f008:**
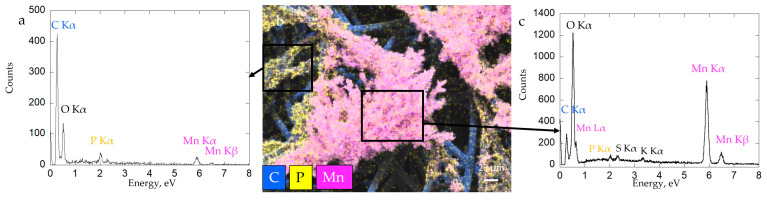
SEM-XEDS analysis of Biofilm-MnO_x_: SEM-EDX cartography with elemental maps of Mn, P and C (**b**) and XEDS spectrum of analyzed zones (black spares) correspond with biofilm (**a**) and biominerals (**c**).

**Figure 9 microorganisms-11-00603-f009:**
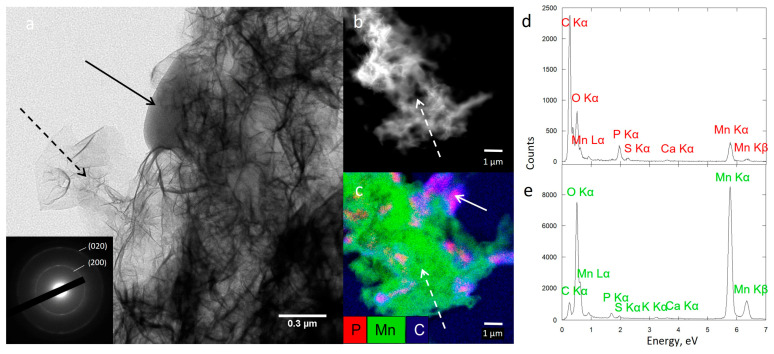
TEM analysis of Biofilm-MnO_x_ (**a**) showing bacteria (solid arrow) and extracellular biominerals (dotted arrow). Electron diffraction in the TEM (inset in (**a**)) is consistent with birnessite. STEM image showing extracellular biominerals (dotted arrow) (**b**) and STEM-XEDS composite elemental map of Mn, C and P with bacteria (solid arrow) (**c**). XEDS spectrum of one bacterium (**d**) and biominerals (**e**).

**Figure 10 microorganisms-11-00603-f010:**
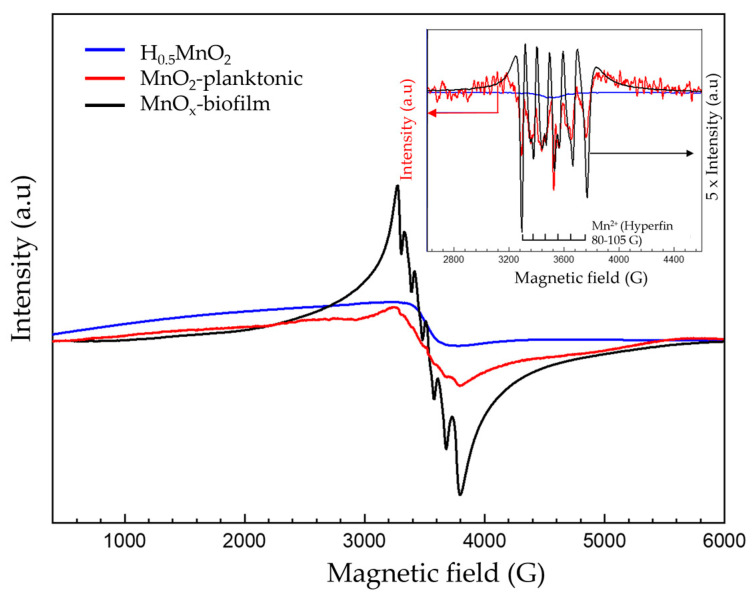
Room temperature EPR spectrum; inset: pseudomodulation of biofilm-MnO_x_ (black), planktonic-MnO_2_ (red) and H_0.5_MnO_2_ (blue). Experimental conditions: power: 2 mW; amplitude modulation: 2 gauss.

**Figure 11 microorganisms-11-00603-f011:**
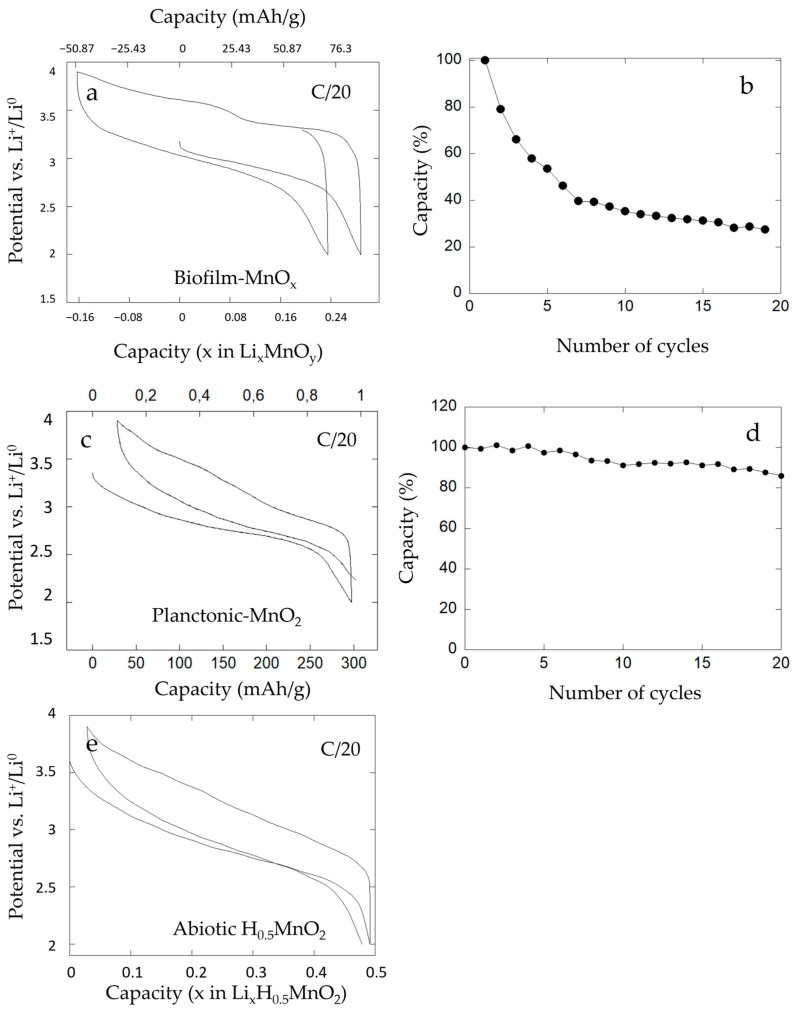
Electrochemical cycling of (**a**) biofilm-MnO_x,_ (**c**) planktonic-MnO_2_ and (**e**) abiotic H_0.5_MnO_2_ vs. Li°. Planktonic-MnO_2_ and abiotic H_0.5_MnO_2_ were tested with the addition of black carbon. Galvanostatic cycling curves were obtained at C/20 (1 Li exchanged in 20 h) starting at discharge. Capacity retention was in charge of (**b**) biofilm-MnO_x_ and (**d**) planktonic-MnO_2_ at C/20 (1 Li in 20 h) for the first twenty cycles.

**Figure 12 microorganisms-11-00603-f012:**
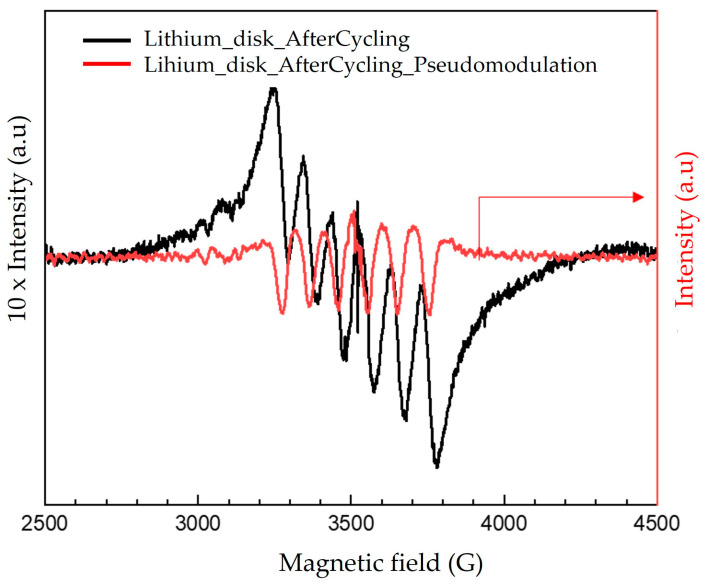
CW EPR spectrum together with pseudomodulation of Li–disk–water solution obtained after one cycle. Experimental conditions: power: 10 mW; amplitude modulation: 2 gauss.

**Table 1 microorganisms-11-00603-t001:** Composition of the rich and mineralization media used for bacterial cultures.

Rich Medium		Mineralization Medium	
Composition	Concentration	Composition	Concentration
Beef extract	3 g/L	HEPES	10 mM
Peptone	5 g/L	(NH_4_)_2_SO_4_	2 mM
MnSO_4_·H_2_O	50 µM	NaCl	0.7 mM
		Glucose	1 mM
		MnSO_4_·H_2_O	0.2 mM

## Data Availability

All datasets generated for this study are included in the article.

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
