# Peer review of "Biologically Assisted One-Step Synthesis of Electrode Materials for Li-Ion Batteries"

_microorganisms, 2023, doi:10.3390/microorganisms11030603_

Round 1

Reviewer 1 Report

Title: Biologically assisted one-step synthesis of electrode materials for Li-ion batteries

Journal: Microorganisms (ISSN 2076-2607)

Manuscript ID: microorganisms-2203628

After carefully evaluation. I am pleased to send you some comments. Please consider these suggestions as listed below to prepare the article again.  

  1. The title seems ok.
  2. The abstract seems very OK. Please add introductory lines in the beginning.
  3. Keywords are ok
  4. Research gap should be delivered on more clear way with directed necessity for the future research work.
  5. Introduction section must be written on more quality way, i.e., more up-to-date references addressed.
  6. The novelty of the work must be clearly addressed and discussed, compare previous research with existing research findings and highlight novelty.
  7. What is the main challenge?
  8. What is problem statement? State clearly.
  9. Do not use lumpy references. Maximum should be 2 or 3. Please revise your paper accordingly since some issue occurs on several spots in the paper.
  10. Please check the abbreviations of words throughout the article. All should be consistent.
  11. The main objective of the work must be written on the clearer and more concise way at the end of introduction section.
  12. The introduction is completely massive.
  13. There are several irrelevant reference please take a strong revision in this section.
  14. Please provide space between number and units. Please revise your paper accordingly since some issue occurs on several spots in the paper.
  15. What is about data repeatability?
  16.  Each section of results does not have scientific output why?
  17. To meet the journal standard, author should add a comparative profile.
  18. Author should add a separate section with name challenges and future perfectives.
  19. Conclusion section is missing some perspective related to the future research work, quantify main research findings, and highlight relevance of the work with respect to the field aspect.
  20. To avoid grammar and linguistic mistakes, MAJOR level English language should be thoroughly checked. Please revise your paper accordingly since several language issue occurs on several spots in the paper.
  21. Reference formatting need carefully revision. All must be consistent in one format. Please follow the journal guidelines.  
  22. Article formatting is weird.

Decision = Major revision. It was tough for me to read and go through, but I was able to make a comments for improvement. Please put forth your best efforts and revised it. The idea is good so, I recommend a chance to revise it.

Author Response

  1. The title seems ok.
  2. The abstract seems very OK. Please add introductory lines in the beginning.
  3. Keywords are ok
  4. Research gap should be delivered on more clear way with directed necessity for the future research work.

The research gap is that all electrode synthesis so far required some post-processing after biomineralization such as heating and adding a polymer binder or conductive carbon to obtain good electrochemical performances. This is now clearly explained in the revised introduction according to the request of referee 1.

  1. Introduction section must be written on more quality way, i.e., more up-to-date references addressed.

The introduction section was corrected accordingly.

  1. The novelty of the work must be clearly addressed and discussed, compare previous research with existing research findings and highlight novelty.

The introduction section was corrected accordingly.

  1. What is the main challenge?

The main challenge of this study is to limit the post-synthesis processing steps and produce an electrode material directly on a collector. This is now stated clearly in the introduction.

  1. What is problem statement? State clearly.

The main problem of these syntheses is that they so far required some post-processing after biomineralization such as heating and adding a polymer binder or conductive carbon to obtain good electrochemical performances. This is now clearly explained in the introduction.

  1. Do not use lumpy references. Maximum should be 2 or 3. Please revise your paper accordingly since some issue occurs on several spots in the paper.

The introduction section was corrected accordingly.

  1. Please check the abbreviations of words throughout the article. All should be consistent.

Abbreviations of words were corrected accordingly.

  1. The main objective of the work must be written on the clearer and more concise way at the end of introduction section.

Our objective is to propose an alternative way of direct electrode synthesis. This is now clearly explained in the introduction.

  1. The introduction is completely massive.

The introduction section was corrected accordingly.

  1. There are several irrelevant reference please take a strong revision in this section.

The introduction section was corrected accordingly.

  1. Please provide space between number and units. Please revise your paper accordingly since some issue occurs on several spots in the paper.

The article was corrected accordingly.

  1. What is about data repeatability?

The synthesis of the material is always reproducible; at the electrochemical level we could obtain similar profiles several times. Experiments were repeated at least three times. This answer is now specified in revised text in method sections.

  1.  Each section of results does not have scientific output why?

Scientific outputs were added in each section of results accordingly.

  1. To meet the journal standard, author should add a comparative profile.

We did not fully understand this request. We suppose it deals with providing more comparisons with analog studies. References to similar studies were provided in the introduction. This discussion requested by the referee is now clearly present in the results section (output 3.3).

  1. Author should add a separate section with name challenges and future perfectives.

A section was added at the end of the conclusion which now stands: “Our method, using a one-pot production at room temperature with a biofilm leads to improve electrochemical performances of a birnessite-based electrode. In future work, the challenge would be to optimize the MnO2/organic proportions and Mn2+ biological oxidation efficiency in order to decrease Mn2+ remnants in the final material so as to obtain a more efficient electrode material without the presence of extra capacity. The possibility to obtain an optimized texture of the electrode material without extra capacity could further ensure a better exchange with lithium ions. One solution to optimize the performance of this material could be to use voltage polarisation of the current collector during biofilm formation. Polarization can have a favorable effect on the colonization of the biofilm on a support or a repulsive effect depending on the bacterial strain, the hydrophilic or hydrophobic nature of the support, or the different ionic forces at play in the culture medium [70,71]. The possibility to optimize the colonization of biofilm on the current collector with an improved part of active matter could be an efficient way of improving manganese oxide production. The facility to obtain microscale textured manganese oxides directly on the current collector could pave the way for more rational production of electrode materials.

  1. Conclusion section is missing some perspective related to the future research work, quantify main research findings, and highlight relevance of the work with respect to the field aspect.

See answer 18

  1. To avoid grammar and linguistic mistakes, MAJOR level English language should be thoroughly checked. Please revise your paper accordingly since several language issue occurs on several spots in the paper.

We asked a high-level English-speaking and writing colleague to correct the revised paper.

  1. Reference formatting need carefully revision. All must be consistent in one format. Please follow the journal guidelines.  

This was done accordingly.  

  1. Article formatting is weird.

Article was corrected accordingly

Decision = Major revision. It was tough for me to read and go through, but I was able to make a comments for improvement. Please put forth your best efforts and revised it. The idea is good so, I recommend a chance to revise it. We have implemented all the changes required by referee 1 and hope that the revised version now meets her/his expectations.

Reviewer 2 Report

Laura Galezowski and co-workers used the bacterium, Pseudomonas putida strain MnB1 to produce manganese oxide biofilm, and then characterized the biofilm-mineral assembly using a combination of technologies. Moreover, the authors applied the biofilm-mineral assembly as the electrode materials for Li-ion batteries. The innovative significance of this study is that it provides insights into the properties of biomineralized biofilms and their possible use to design new one-pot electrode synthesis pathways. The whole manuscript is organised logically and hihgly interesting to researchers in the filed of electrode materials in Li-ion batteries. Therefore, i recommended it to be published in Microorganisms after resolving the following questions. And the detailed questions are listed as following:

1) The signal (4000 g) of rotation in line 152 of page 4 should be incorrect, and should replace with “4000 r”.

2) In line 153-154 of page 4, the authors claimed that “Colonized GDL were collected and immersed 3 times in water for 10 minutes. Samples were further dried under air at 80 °C for two hours”. Can the water be completely removed from the sample after dried under air at 80 °C for two hours? If the water is present in the sample, would the relevant characterization be affected?

3) Is the biomineral physically or chemically adsorbed upon the fibers of current collector? And how about the strength of the biomineral adsorbed upon the fibers of current collector?

4) The authors should briefly describe the Li-ion batteries to highlight the importance of the research and then other published articles should be cited in the introduction (Nano Energy, 2020, 76, 104964; Solar RRL, 2021, 6(4), 2100839; ACS Applied Materials & Interfaces, 2022, 14, 41389-41399, et al).

5) The language in the manuscript should be carefully checked and improved.

Author Response

  • The signal (4000 g) of rotation in line 152 of page 4 should be incorrect, and should replace with “4000 r”.

The abbreviation was corrected to “4000 x g”. 4000 x g of rotation is the conversion of the centrifuge rotor speed (RPM). RPM depends on the centrifuge (rotor radius from center of the rotor to sample) whereas RCF (units of gravity, x g) is independent of the centrifuge used.  This is now clarified in the text in the methods section.  

2) In line 153-154 of page 4, the authors claimed that “Colonized GDL were collected and immersed 3 times in water for 10 minutes. Samples were further dried under air at 80 °C for two hours”. Can the water be completely removed from the sample after dried under air at 80 °C for two hours? If the water is present in the sample, would the relevant characterization be affected?

After drying samples under air at 80 °C for two hours, the surface water can be completely removed. This is shown by the fact that a fraction of water has left the sample at or below this temperature as shown by TGA analysis (Figure 6). Moreover, if such surface water was present, lithium exchange would be affected. Then water released above 80°C up to 200 °C as reported on Thermal analyses (Figure 6) might affect interlayer cations. We thus suppose that interlayers in birnessite are not significantly affected below 80°C and that all water loss in this temperature range concerns surface water only. This is now clarified in the revised version of the text in the method section.

3) Is the biomineral physically or chemically adsorbed upon the fibers of current collector? And how about the strength of the biomineral adsorbed upon the fibers of current collector?

First, bacteria are able to produce a biofilm and to colonize the surface of fibers thanks to extracellular polymeric substances (natural polymers of high molecular weight). Then the active MnO2 mineral is developing on the surface of the biofilm but is intimately engulfed in the organic matter (see figure 8). The question of the referee is thus about the strength of adhesion of biofilms on substrates which is a well-studied topic (see for example references biofouling which provides indications about the adhesion of our composite material on the carbon current collector). This discussion requested by the referee is now clearly present in the results section (output 3.2).

4) The authors should briefly describe the Li-ion batteries to highlight the importance of the research and then other published articles should be cited in the introduction (Nano Energy, 202076, 104964; Solar RRL, 20216(4), 2100839; ACS Applied Materials & Interfaces, 202214, 41389-41399, et al).

Modifications and others articles were cited in the introduction.

5) The language in the manuscript should be carefully checked and improved.

We asked a high-level English-speaking and writing colleague to correct the revised paper.

Reviewer 3 Report

Authors have done the good work to present this manuscript. But there are some comments and need to be addressed.

1. Author should put composite sample XRD spectrum for comparison before and after.

2. Author should provide nitrogen adsorption data and spectrum for evidence for mesoporous.

3. Author should provide the power rate experiments for comparison texture and untextured samples.

4. Page 4, line 164, there should be space between in and 100 mL.

5. Authors should go through the once again the manuscript to fix the minor issues like space, prepositions and some grammatical errors.

Author Response

  1. Author should put composite sample XRD spectrum for comparison before and after.

The carbon electrode does not allow us to do an XDR after one cycle because the amount of amorphous material (carbon of current collector) is too important compare to mineral part. One word was added about that in the methods section.

  1. Author should provide nitrogen adsorption data and spectrum for evidence for mesoporous.

In the same way, it takes a lot of material to have a minimum surface of 20m2 necessary for the surface analysis. We have already tried to do this analysis without success because we do not have the necessary quantities. Now in the text in the methods section.

  1. Author should provide the power rate experiments for comparison texture and untextured samples.

The carbon layer (current collector) prevents us from making these power rate measurements with the ground biofilm. Since our material is on a carbon current collector, we cannot unhook the powder to grind it because we don't have enough material. Explained in the revised version in the methods section.

  1. Page 4, line 164, there should be space between in and 100 mL.

This was done in the revised version.

  1. Authors should go through the once again the manuscript to fix the minor issues like space, prepositions and some grammatical errors.

We asked a high-level English-speaking and writing colleague to correct the revised paper.

Round 2

Reviewer 1 Report

Accepted in the present form.

Reviewer 2 Report

Accept in present form

Reviewer 3 Report

Please accept the manuscript in present form. Author provided enough details.